# Acid Sphingomyelinase Activation and ROS Generation Potentiate Antiproliferative Effects of Mitomycin in HCC

**DOI:** 10.3390/ijms252212175

**Published:** 2024-11-13

**Authors:** Sirkka Buitkamp, Stephanie Schwalm, Katja Jakobi, Nerea Ferreiros, Christin Wünsche, Stefan Zeuzem, Erich Gulbins, Christoph Sarrazin, Josef Pfeilschifter, Georgios Grammatikos

**Affiliations:** 1Pharmazentrum Frankfurt, Institut für Allgemeine Pharmakologie und Toxikologie, Goethe University Hospital, 60590 Frankfurt am Main, Germany; sirkkaschroeter@hotmail.de (S.B.); s.schwalm@med.uni-frankfurt.de (S.S.); jakobi@med.uni-frankfurt.de (K.J.); christin.wuensche@web.de (C.W.); pfeilschifter@em.uni-frankfurt.de (J.P.); 2Klinik für Innere Medizin I, Helios Dr. Horst Schmidt Kliniken Wiesbaden, 65199 Wiesbaden, Germany; 3Medizinische Klinik 2/Rheumatologie, Goethe University Hospital, 60590 Frankfurt am Main, Germany; 4Pharmazentrum Frankfurt, Institut für klinische Pharmakologie, Goethe University Hospital, 60590 Frankfurt am Main, Germany; ferreirosbouzas@em.uni-frankfurt.de; 5Multidos, 65520 Bad Camberg, Germany; 6GBG Forschungs GmbH, 63263 Neu-Isenburg, Germany; 7Medizinische Klinik 1/Gastroenterologie und Hepatologie, Goethe University Hospital, 60590 Frankfurt am Main, Germany; zeuzem@em.uni-frankfurt.de; 8Institute of Molecular Biology, University Hospital Essen, University of Duisburg-Essen, 45122 Essen, Germany; erich.gulbins@uni-due.de; 9Medizinische Klinik II, St. Josefs-Hospital Wiesbaden, 65189 Wiesbaden, Germany; csarrazin@joho.de; 10St’ Lukes Hospital, 55236 Thessaloniki, Greece

**Keywords:** ASM, ceramide, reactive oxygen species, sphingolipids, hepatocellular carcinoma

## Abstract

Sphingolipids play a major role in the regulation of hepatocellular apoptosis and proliferation. We have previously identified sphingolipid metabolites as biomarkers of chronic liver disease and hepatocellular carcinoma. Human hepatocellular carcinoma cell lines were transfected with a plasmid vector encoding for acid sphingomyelinase. Overexpressing cells were subsequently treated with mitomycin and cell proliferation, acid sphingomyelinase activity, sphingolipid concentrations, and generation of reactive oxygen species were assessed. The stimulation of acid sphingomyelinase-overexpressing cell lines with mitomycin showed a significant activation of the enzyme (*p* < 0.001) followed by an accumulation of various ceramide species (*p* < 0.001) and reactive oxygen radicals (*p* < 0.001) as compared to control transfected cells. Consequently, a significant reduction in cell proliferation was observed in acid sphingomyelinase-overexpressing cells (*p* < 0.05) which could be diminished by the simultaneous application of antioxidant agents. Moreover, the application of mitomycin induced significant alterations in mRNA expression levels of ceramidases and sphingosine kinases (*p* < 0.05). Our data suggest that the overexpression of the acid sphingomyelinase in human hepatoma cell lines enhances the *in vitro* antiproliferative potential of mitomycin via accumulation of ceramide and reactive oxygen species. The selective activation of acid sphingomyelinase might offer a novel therapeutic approach in the treatment of hepatocellular carcinoma.

## 1. Introduction

Hepatocellular carcinoma (HCC) represents the most common primary liver cancer and the second leading cause of tumor-related mortality with an alarming rising incidence and mortality worldwide [1,2]. About 90% of HCC develops on a cirrhotic background while chronic infection with hepatitis B (HBV) or hepatitis C virus (HCV), chronic alcohol abuse and nonalcoholic fatty liver disease (NAFLD) constitute the most common risk factors [3,4]. Independently from the etiology, it is estimated that about 30% of cirrhotic patients will develop HCC during their lifetime [5].

Unfortunately, HCC is in most cases diagnosed at an advanced stage, in which potentially curative treatment options such as surgery, percutaneous ablation, and liver transplantation are no longer available [6]. Since the initial approvement of sorafenib, a multi-kinase inhibitor [6], many novel immune-therapeutic regimes have significantly updated the systemic treatment options of HCC [7], though confined only in patients with advanced disease and no therapeutic perspective. Thus, transarterial chemoembolization (TACE) of HCC serves as a bridging treatment of patients with HCC eligible for liver transplantation or as a palliative treatment option in patients with advanced HCC [8] and, according to recent promising data, even in combination with immune-therapy as a stimulator of the immune response [9]. However, the shortage of available liver allografts and limited efficacy of systemic treatment urge the identification of novel targets and approaches in HCC therapy.

Sphingolipids (SLs) constitute bioactive signaling molecules that modulate fundamental cellular processes [10] thus playing an important role in the pathogenesis of various diseases including atherosclerosis, bacterial and viral infections, obesity, insulin resistance, and SL-storing disorders called sphingolipidoses [11]. Particularly in cancer pathogenesis, SL metabolism has lately attracted great interest since SLs regulate proliferation and apoptosis of tumors and thus chemotherapeutic drug resistance [12]. Furthermore, SLs play a critical role in the regulation of injury, repair, and regeneration of the liver [13] and thereby in the pathophysiology of various liver diseases [14]. Ceramide (Cer), the hydrophobic backbone of various complex SLs and mediator of antiproliferative responses, was found to exert hepatotoxic effects by stimulation of hepatic inflammation [15]. Either by *de novo* synthesis or by hydrolysis of sphingomyelin via action of acid (ASM) or neutral sphingomyelinase, ceramide accumulates in response to several stress stimuli including hypoxia, ischemia, pro-inflammatory cytokines TNFα, and IL-1ß as well as death receptor activation [13]. Ceramide generation is associated with the formation of reactive oxygen species (ROS), direct mitochondrial toxicity and inhibition of anti-apoptotic signal transducers, thus inducing apoptosis [15]. Sphingosine-1 phosphate (S1P), the functional antagonist of ceramide stimulating proliferation and cell survival, is also proposed to regulate hepatocellular susceptibility to various stimuli [14,16] and hepatocarcinogenesis both *in vitro* and *in vivo* [17,18]. In this context, both *in vitro* and *in vivo* data have shown so far that a lack of ASM or its pharmacological inhibition protects cells against hepatotoxic effects [15], while the addition of recombinant ASM augmented the antiproliferative effects of sorafenib treatment in a HCC mouse model [19]. However, the role of ASM as a potential target in HCC therapy still remains largely unexplored.

We have previously shown that the overexpression of ASM sensitized glioma cells to cytostatic therapy [20] while chronic hepatitis C infection and non-alcoholic fatty liver disease is associated with an increased serologic ASM activity [21]. Further data from our group revealed an up-regulation of C16-ceramide and S1P in the serum of HCC patients and proposed in this way a role of SLs as potential serum biomarkers of HCC [22] while serum C16-ceramide was able to independently predict de novo HCC in cirrhotic patients after the successful eradication of HCV [23]. The purpose of the current study is thus to investigate whether the overexpression of ASM sensitizes HCC cells to chemotherapy with mitomycin C (MMC), which is usually used within TACE-regimens in the clinical setting. We further studied the impact of MMC treatment on SL metabolism and the generation of ROS *in vitro*.

## 2. Results

### 2.1. Mitomycin Activates Acid Sphingomyelinase in HCC Cell Lines

In our study, we initially transfected the HCC cell lines Huh7.5 and HEPG2 with a plasmid vector encoding for ASM (pEF-ASM) and generated cell clones with a stable overexpression of the enzyme, while control cell clones were transfected with the empty vector (pEF-pJK). Transfection resulted in a 10-fold increase in mRNA expression of ASM in Huh7.5 cells and a 3-fold increase in the HepG2 cell line (Figure 1A). Consequently, ASM activity was up-regulated significantly as compared to cell clones transfected with the empty vector (Figure 1B). We focused on the effect of mitomycin, which is usually included in locally applied chemotherapy regimens in patients with HCC, on the sphingolipid metabolism and especially on the expression and activity of ASM. As shown in Figure 1D, the stimulation of transfected Huh7.5 and HepG2 cells with mitomycin led to a significant upregulation of ASM activity, due to a significant upregulation of ASM mRNA expression. Upregulation of mRNA expression was more prominent in cells transfected with the control vector (pEF-pJK, Figure 1C) since ASM-overexpressing cells (pEF-ASM) showed a higher baseline expression of ASM-mRNA due to respective transfection.

### 2.2. Ceramide Levels Upregulated upon Stimulation with MITOMYCIN

Since ceramide constitutes the enzymatic product of ASM, we performed quantification via LC-MS of several ceramide subspecies upon treatment with mitomycin. This revealed a significant cellular accumulation of all ceramide subspecies in Huh7.5 cells with the increase being more prominent in ASM-overexpressing cells. On the contrary, in HepG2 cells, we solely observed a significant increase in C24-ceramide levels. Further ceramide subspecies were not significantly elevated showing solely a positive trend in ASM-overexpressing cells (Figure 2).

### 2.3. Mitomycin Induces Cell Death via ASM Activation and ROS Generation

As a next step, we investigated the effect of mitomycin-mediated upregulation of ASM on the induction of cell death. The treatment of Huh7.5 and HepG2 cells with mitomycin resulted in a significant decrease in cell viability (Figure 3A). The overexpression of ASM sensitized HCC cell lines even further as compared to control transfected cells thus enhancing the antiproliferative effects of MMC. In order to further validate the role of ASM, imipramine, a tricyclic antidepressant drug and known pharmacological inhibitor of ASM [24], was co-administered. Accordingly, the effect of MMC on cell death was reduced in Huh7.5 and HepG2 cells. Furthermore, we observed that the antiproliferative effects of mitomycin are based on the production of ROS. We identified a significant increase in ROS in ASM-overexpressing Huh7.5 cells and a positive trend in control transfected cells, whereas in HepG2 cells no significant change could be observed (Figure 3B). Treatment with the antioxidative agents Tiron and N-acetyl-cysteine confirmed the observations, reducing the antiproliferative effect of mitomycin (Figure 3C).

### 2.4. Stimulation with Mitomycin Induces Deregulation of Sphingolipid Metabolism

As the sphingolipid metabolism is a dynamic poised process, we tested whether mitomycin induced compensatory effects in addition to the activation of ASM and ceramide accumulation. Quantifications via LC-MS of lipid extracts upon treatment with MMC revealed a significant increase in cellular levels of sphingosine, S1P, and sphinganine in both Huh7.5 and HepG2 cells. The effect was more prominent in ASM-overexpressing cells (Figure 4). Furthermore, in Huh7.5 cells, mitomycin triggered the up-regulation of acid and neutral ceramidase with the effect being more distinct in the ASM transfected cells, as shown in Figure 5. In HepG2 cells, an increased mRNA expression of acid ceramidase was induced as well, although the basal expression of the neutral ceramidase was significantly reduced. Additionally, mRNA expression of S1P-phosphatase was upregulated in Huh7.5 cells as well, whereas no significant change in HepG2 cells was observed. The observed accumulation of S1P strongly suggested an up-regulation of sphingosine kinases (SK). In fact, exposure to mitomycin lead to significant enhanced mRNA synthesis of SK-1 and SK-2 in the control-transfected Huh7.5 cells beside a positive trend in ASM-transfected cells (Figure 5). In HepG2 cells, mRNA expression of SK-2 was significantly increased, while SK-1 steady state level was significantly reduced (Figure 5).

## 3. Discussion

The accurate assessment of various bioactive SL metabolites via mass spectrometry has enabled the investigation of their role in various aspects of translational medicine. We have previously observed significant variations in SL metabolites in the serum of patients with chronic liver disease [21] and identified SLs as biomarkers of liver fibrosis [25] and HCC [22,23]. In the present study, we observed that the modulation of SL signaling, as induced by the overexpression of ASM, is able to sensitize human HCC cell lines to mitomycin and could offer a potential novel strategy in optimizing HCC treatment.

Our data illustrate that administration of the chemotherapeutic agent mitomycin in HCC cell lines results in a significant up-regulation of ASM activity (Figure 1), hence leading to a considerable accumulation of pro-apoptotic ceramide (Figure 2). Mitomycin appeared to act with a high variability regarding the induced enzymatic effects, the accumulation of ceramides, and the consecutive change in cell proliferation and viability as illustrated by the high standard deviation observed in Figure 1, Figure 2, Figure 3, Figure 4 and Figure 5. This could be partially attributed to an enhanced cytotoxicity of mitomycin especially in ASM-transfected cells, as the overexpression of ASM appeared to “oversensitize” cell lines to the agent. In the meantime, it is commonly accepted that ASM regulates substantially cellular responses to stress factors. In addition to various chemotherapeutic drugs like doxorubicin or gemcitabine, ASM has been further shown to be activated by other stress stimuli such as ionizing and ultraviolet (UV) radiation, death receptor signaling [10,26], or the generation of ROS [27]. ASM deficiency reverses apoptosis induction, pointing out its critical role on the regulation of cellular sensitivity to stress stimuli [26,28,29,30]. This is consistent with our current observations that the inhibition of ASM by imipramine diminishes the apoptotic effect of mitomycin in human HCC cell lines (Figure 3). Thus, we may hypothesize that the upregulation of ASM activity might confer the sensitivity of HCC towards locoregional antiproliferative treatments, such as mitomycin, with the latter being already clinically in use within TACE formulations. Yet, at present, it is not completely clarified how ASM activation is being mediated. Activation of the enzyme has been associated with the production of ROS [10] while scavengers of ROS intermediates are able to inhibit the stimulation of the enzyme [10,20]. Our current data show a marked elevation in ROS generation upon mitomycin stimulation while the co-administration of Tiron, a ROS scavenger, diminished the antiproliferative effects of the medication dependent on ASM activity. Therefore, our results suggest in conclusion that the antiproliferative effects of mitomycin are induced by the generation of ROS, consecutive ASM activation, and the accumulation of ceramide.

Beside the observed activation of ASM, it remains intriguing how the expression of the enzyme is able to regulate the sensitivity of HCC towards chemotherapy. Interestingly, recent data from the Oncomine database revealed a significant downregulation of the sphingomyelin phosphodiesterase 1 (SMPD1) gene, which encodes for ASM, in HCC as compared to normal liver tissue [19]. On the contrary, ASM is upregulated in chronic liver inflammation [21], supporting the assumption that ASM deregulation is an important mechanism in the pathogenesis of chronic liver disease and HCC. In the current study, we were, to our knowledge, the first to demonstrate that the overexpression of ASM sensitizes HCC cells for the administration of mitomycin by enhancing its antiproliferative effects (Figure 3). Similar observations were previously made in glioma, where ASM-overexpression sensitized the tumor cells to doxorubicin and gemcitabine [20]. Also, in ovarian cells and mouse melanomas, overexpression of the enzyme was shown to sensitize cells to irradiation [31]. Moreover, the higher expression of neutral sphingomyelinase, another member of the family of SMPD, was further shown to reduce proliferation in HCC cell lines whereas its knockdown facilitated tumor growth *in vitro* and *in vivo* [32]. Additionally, Savić et al. showed that the application of recombinant ASM (rhASM), typically used for enzyme replacement therapy in Niemann-Pick Disease, as an adjuvant to sorafenib resulted in a significant reduction in tumor proliferation in the murine HCC model [19]. This corresponds with our current observations that the antiproliferative effect of sorafenib in HCC cells is potentiated by the overexpression of the enzyme. In summary, our data implicate that the modulation of the SL metabolism by targeting ASM may potentially provide an additional advance to common cancer strategies.

The generation of ceramide constitutes a key concept in tumor suppression. Our data identified a marked accumulation of intracellular ceramides with distinct acyl chain lengths upon stimulation with MMC, with the effect being augmented in ASM overexpressing cells (Figure 2). However, it is well described in the literature that not every ceramide species as defined by their chain length appears to have the same second messenger effect. Recent studies indicated a chain length dependency of distinct ceramides in proliferation [12,33,34]. Long chain ceramides (C16–C20) show antiproliferative effects, while very long chain ceramides (>C24) seem to promote proliferation [33]. In our current study, both long and very long chain ceramides showed an accumulation in HCC cells consequently to ASM activation by MMC (Figure 2). While ceramides accumulate in Huh7.5 cells upon stimulation with MMC, in the HepG2 control transfected cells, ceramide concentrations even decreased as compared to the ASM-overexpressing cells. Both the concomitant upregulation of very-long chain ceramides as well as the distinct ceramide concentrations among both cell lines may reflect compensatory effects within the SL metabolism and different mRNA-expression patterns of SL regulating enzymes upon antiproliferative treatment (please see Figure 2 and Figure 5). As known so far, activation of the acid sphingomyelinase is able to cleave sphingomyelins with distinct acyl chain lengths and thus leads to significant changes in ceramide concentrations in a similar but not equal manner regarding the concentrations of produced ceramides as also observed in our study (Figure 2). This becomes further apparent as we observed an increase in the concentrations of sphingosine (Sph), S1P, and sphinganine 1–P upon mitomycin treatment (Figure 4). Since both S-1-P and Sa-1-P are increased and do not really differ between pJK (control-) and ASM-transfected cells, we assume that mitomycin also blocks the activity of the lyase that mediates the consumption of S-1-P and Sa-1-P. Furthermore, neutral (nCase) and acid ceramidase (aCase), enzymes segregating ceramide to sphingosine, are upregulated in Huh 7.5 cells after stimulation with mitomycin, while in HepG2 cells only the aCase is upregulated (Figure 5). It might seem paradoxical at first sight that S1P is upregulated despite decreased cell proliferation upon ASM overexpression and activation via mitomycin, but we assume it as a compensatory response of the cell within SL signaling after accumulation of mitomycin. This is consistent with the fact that S1P-phosphatase is activated by mitomycin, at least in Huh7.5 cells (Figure 5). Interestingly, significant lower mRNA expression levels of the S1P phosphatase 1 (SGPP1) in HCC tissue have been identified as compared to normal livers [19]. Thus, both our current observations as well as data in the literature suggest a modulatory role of ceramide-catabolizing enzymes in hepatocarcinogenesis and probably HCC susceptibility to chemotherapy.

In summary, our data show that mitomycin activates ASM via ROS generation hence leading to the accumulation of ceramide. Despite the compensatory changes in SL regulating enzymes with the consecutive upregulation of pro-proliferative SL metabolites, the overexpression of ASM sensitizes HCC tumor cells to chemotherapy with mitomycin. Targeting of the SL metabolism may offer a novel therapeutic approach in HCC pathophysiology. Further studies are needed in order to elucidate the therapeutic potential of SL signaling in HCC.

## 4. Materials and Methods

### 4.1. Cells

The human HCC cell lines Huh7.5 and HEPG2 were cultured in D-MEM medium supplemented with 10% FCS, 10 mM HEPES, 2 mM L-glutamine, 1 mM sodium pyruvate, 100 mM nonessential amino acids, 100 units/mL penicillin, and 100 μg/mL streptomycin (all purchased from GIBCO/BRL-Life Technologies, Karlsruhe, Germany). The cells were stably transfected with an expression vector for acid sphingomyelinase (pEF-ASM) [20], inducing a strong expression of acid sphingomyelinase. The control cells were transfected with the empty vector (pEF) [20]. The cells were treated with mitomycin for the indicated time. If indicated, the cells were pretreated with 1 mM Tiron for 10 min prior to the addition of the chemotherapeutic drugs.

### 4.2. Cell Proliferation and ROS Formation

Cell viability was measured using alamarBlue reagent (resazurin) (Life Technologies, Thermo Fisher Scientific, Waltham, MA, USA) [20]. The cells were plated in a black 96-well plate (15 × 10^3^ cells per well) and maintained overnight in complete medium. The cells were then treated as indicated and for the last 4 h, alamarBlue reagent was added. Intensity of fluorescence reflecting cell viability was measured at 544/590 nm excitation/emission wavelengths with a SpectraMax microplate reader (Molecular Devices, LLC, Sunnyvale, CA, USA). The generation of reactive oxygen species (ROS) was determined by using a specific ROS-activity assay KIT (ENZO Total ROS/Superoxide Detection Kit, Enzo Life Sciences, Farmingdale, NY, USA) as described in the manufacturer’s manual.

### 4.3. Acid Sphingomyelinase Activity

To determine the activity of acid sphingomyelinase, cells were treated with 3 µM MMC for the indicated time in cell culture medium. Higher doses of the drugs were used for measuring the activity of acid sphingomyelinase and ceramide levels than for the induction of cell death to obtain a synchronized cell response in these assays. The medium was removed, the cells were lysed in 250 mM sodium acetate (pH 5.0) and 1% NP-40 for 10 min on ice, the samples were diluted to 0.1% NP-40 and 250 mM sodium acetate (pH 5.0), brought to a volume of 300 μL, and the enzyme reaction was initiated by the addition of substrate [^14^C] sphingomyelin (52 mCi/mmol; Perkin Elmer, Waltham, MA, USA, #NEC 663010UC). Prior to the addition to cell lysates, the substrate [^14^C] sphingomyelin was dried, resuspended in 250 mM sodium acetate (pH 5.0) and 0.1% NP-40, micelles were formed by a 10 min bath sonication, and an aliquot was added to the cell lysates. The samples were incubated for 30 min at 37 °C, the reaction was stopped by the addition of 800 μL CHCl_3_:CH_3_OH (2:1) and phases were separated by 5 min centrifugation at 14,000 rpm, and the release of [^14^C] phosphorylcholine into the aqueous phase by acid sphingomyelinase activity was determined by liquid scintillation counting of an aliquot of the upper phase. Given the specific activity of [^14^C] Sphingomyelin of 52 mCi/mmol, the activity of 100 cpm/μg/h corresponds to 1 pmol/μg/h specific activity of the acid sphingomyelinase.

### 4.4. Two-Step Polymerase Chain Reaction (PCR) Analysis

Briefly, 1.2 μg of total RNA was isolated with TRIZOL™ reagent (Sigma-Aldrich, Steinheim, Germany) according to the manufacturer’s protocol and used for reverse transcriptase polymerase chain reaction (RT-PCR; RevertAid™ first strand cDNA synthesis kit, Thermo Fisher Scientific, Waltham, MA, USA) utilizing an oligo (dT) primer for amplification. Real-time PCR (TaqMan^®^ Thermo-Fisher Scientific, Waltham, MA, USA) was performed using the Applied Biosystems 7500 Fast Real-Time PCR System. The TaqMan system, all probes, primers, the reporter dyes 6-FAM and VIC, and the Taqman Genotyper software v1.7.1 were from Life Technologies (Darmstadt, Germany). The following TaqMan^®^ assays were used: human sphingosine kinase (SK)-1, Hs00184211_m1; human SK-2, Hs00219999_m1; human S1P-phosphatease 1 (S1PP1), Hs00229266_m1; human neutral ceramidase (nCase), Hs01015660_m1; and human acid ceramidase (aCase), Hs01001653_m1. The cycling conditions were as follows: 95 °C for 15 min (1 cycle), 95 °C for 15 s and 60 °C for 1 min (40 cycles). The threshold cycle (Ct) was calculated by the instrument’s software (7500 Fast System SDS Software version 1.4, Life Technologies, Darmstadt, Germany) Analysis of the relative mRNA expression was performed using the ΔΔCt method. The housekeeping gene human GAPDH (Life Technologies, Darmstadt, Germany) was used for normalization.

### 4.5. Determination of Sphingolipid Concentrations by High-Performance Liquid Chromatography Tandem Mass Spectrometry

For the quantitation of sphingolipids, 20 µL of cell lysates were extracted with methanol:chloroform:HCl (15:83:2). Afterwards, amounts of C16:0-Cer, C18:0-Cer, C20:0-Cer, C24:1-Cer, C24:0-Cer, C16:0-dhCer, C18:0-dhCer, C24:0-dhCer, C24:1-dhCer, and the internal standard C17:0-Cer; and sphingosine, sphingosine-1-phosphate, sphinganine and sphinganine-1-phosphate, and the internal standards (C17-sphingosine and C17-sphingosine1-phosphate) were analyzed by liquid chromatography coupled to tandem mass spectrometry (LC-MS/MS). A Luna C18 column (150 mm × 2 mm ID, 5 µm particle size, 100 Å pore size; Phenomenex, Aschaffenburg, Germany) was used for chromatographic separation. The HPLC mobile phases consisted of water-formic acid (100:0.1, *v*/*v*) (A) and acetonitrile-tetrahydrofuran-formic acid (50:50:0.1, *v*/*v*/*v*) (B). For separation, a gradient program was used at a flow rate of 0.3 mL/min. The initial buffer composition 60% (A)/40% (B) was held for 0.6 min and then in 3.9 min linearly changed to 0% (A)/100% (B) and held for 6.5 min. Subsequently the composition was linearly changed within 0.5 min to 60% (A)/40% (B) and then held for another 4.5 min. The running time for every sample (injection volume is as follows: 15 µL for ceramide and dihydroceramide determination and 10 µL for the other sphingolipids was 16 min. MS/MS analyses were performed on an API4000 (triple quadrupole mass spectrometer, Applied Biosystems, Darmstadt, Germany, Applied Biosystems, Darmstadt, Germany) equipped with an APCI (Atmospheric Pressure Chemical Ionization) ion source (Applied Biosystems, Darmstadt, Germany) for ceramide and dihydroceramide determination, and with an ESI (Electrospray) ion source for sphingosine, sphinganine, and their 1-phosphate derivatives determination. The analysis was conducted in Multiple Reaction Monitoring (MRM) mode. For every analyte, two transitions were recorded: one for quantification and the other for qualification, to exclude false positive results, with a dwell time of 50 ms. For analysis and quantification, the Analyst Software 1.5 (Applied Biosystems, Darmstadt, Germany) was used, and the peak area of the analyte was corrected by the peak area of the internal standard (C17:0-Cer). Linearity of the calibration curve was proven for C16:0-Cer, C24:0-Cer, C16:0-dhCer, C24:1-dhCer, and C24:0-dhCer from 0.6 to 1.000 ng/mL; for C18:0-Cer from 0.18 to 300 ng/mL; for C20:0-Cer and C24:1-Cer from 0.24 to 400 ng/mL; and for C18:0-dhCer from 0.3 to 500 ng/mL. For sphingosine, sphinganine, and their phosphate derivatives, the calibration curve ranged from 0.15 to 250 ng/mL. The coefficient of correlation was at least 0.99. Variations in accuracy were less than 15% over the whole range of calibration. All lysates were stored at −80 °C until assayed. Following the updated nomenclature for MS-derived lipid structures as recently published [35], we refer to the updated definitions since the illustrated abbreviations of lipid metabolites in the current study are in line with traditionally used definitions.

### 4.6. Statistical Analysis

Calculations were made with GraphPad Prism v5.01 for Windows (GraphPad Software, San Diego, CA, USA) and by using the BiAS software for windows (version 9.16, Epsilon-Verlag, Darmstadt, Germany). Statistical comparisons were carried out using the non-parametric Mann–Whitney-U and Kruskal–Wallis tests in order to determine statistically significant differences among patient groups. The data are expressed as Means ± Standard Deviation unless otherwise specified. The level of significance is set at α = 0.05 representing the 95% confidence interval. Statistically significant differences are indicated in the corresponding figures: “*” = *p* < 0.05, “**” = *p* < 0.01, and “***” = *p* < 0.001.

## Figures and Tables

**Figure 1 ijms-25-12175-f001:**
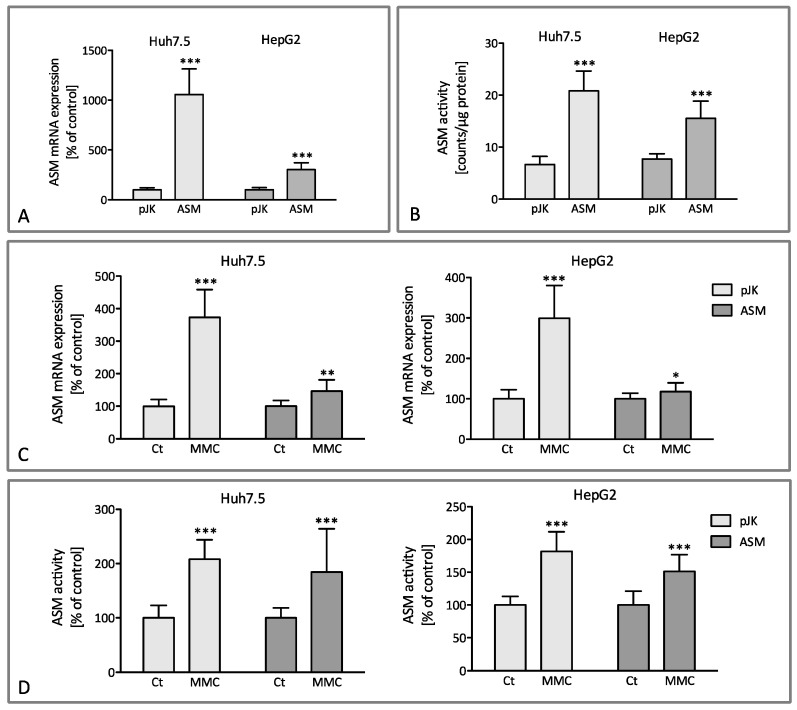
Mitomycin C upregulates the expression and activity of ASM. mRNA-expression and enzyme activity after transfection with either an empty vector (pJK) or a plasmid encoding for ASM (**A**,**B**). The transfected Huh7.5 and HepG2 cells were treated for 48 h either with or without 3 µM mitomycin C (MMC) (**C**,**D**). The data are expressed in % of the basic expression level (**A**) or in % of untreated control (**C**). Given the specific activity of [^14^C] Sphingomyelin of 52 mCi/mmol, the activity of 100 cpm/μg/h corresponds to 1 pmol/μg/h specific activity of the acid sphingomyelinase. Shown are means +/− SD of three independent experiments (*n* = 3). *** *p* < 0.0001, ** *p* < 0.001, and * *p* < 0.05 indicate statistical significance when compared to the control values (**B**,**D**). The results are expressed in counts normalized on extracted protein (**B**) or in % of untreated control (**D**) and are means ± SD (*n* = 3, *** *p* < 0.0001, ** *p* < 0.001, and * *p* < 0.05).

**Figure 2 ijms-25-12175-f002:**
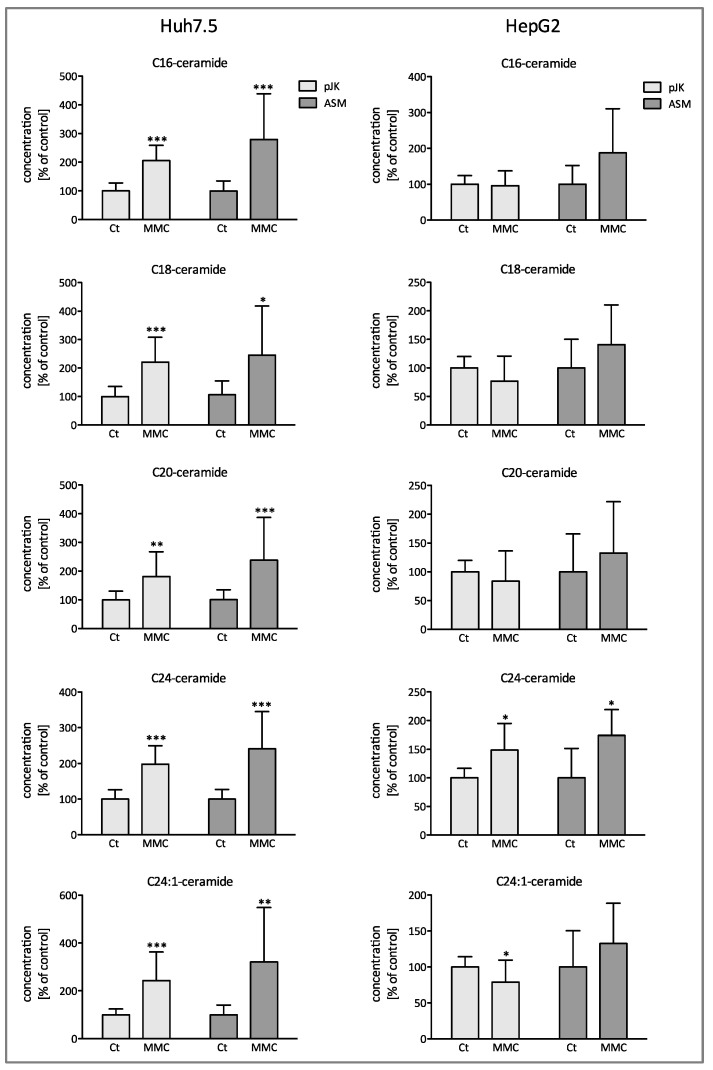
Mitomycin C causes enhanced cleavage of sphingomyeline and upregulates cellular levels of the ceramide species in ASM-overexpressing cells. The ceramide levels in ASM transfected (ASM) and control transfected (pJK) Huh7.5 after treatment with 3 µM mitomycin C (MMC) for 48 h. The data are expressed in % of the untreated control and are means ± SD from a total of three independent experiments (*n* = 3). *** *p* < 0.0001, ** *p* < 0.001, and * *p* < 0.05 are considered significantly different when compared to the control group.

**Figure 3 ijms-25-12175-f003:**
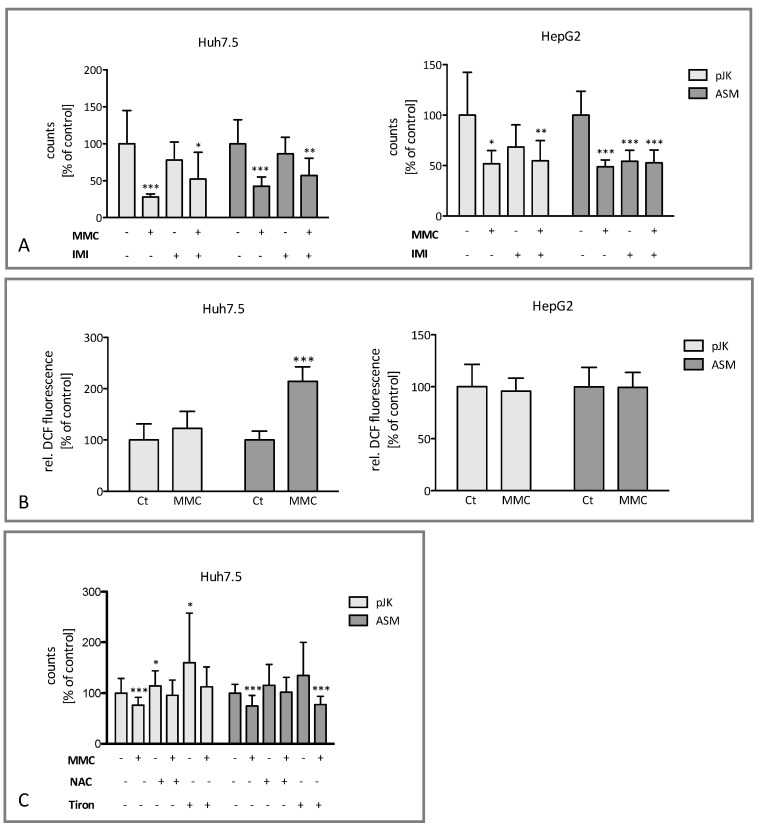
The effect of Mitomycin C on cell viability and ROS production is abrogated by antioxidants. The cells were either pretreated for 1 h with or without 25 µM imipramine prior to stimulation for 48 h either with or without 0.3 µM mitomycin C (MMC) (**A**). The data of cell counts are expressed as % of control and are means ± SD (*n* = 3). *** *p* < 0.0001, ** *p* < 0.001, and * *p* < 0.05 indicate statistical significance when compared to the control group. (**B**) The cells were incubated for 48 h either with or without 0.3 µM MMC C. The data are expressed in % of the untreated control and are means ± SD of at least three (*n* = 3) independent experiments. (**C**) The cells were treated for 48 h with either 0.3 µM MMC C (MMC), 3 mM N-acetyl-Cysteine (NAC), 100 µM Tiron, or combined. The results are expressed as % of control and are means ± SD (*n* = 3). *** *p* < 0.0001, ** *p* < 0.001, and * *p* < 0.05 are significantly different when compared to the control group.

**Figure 4 ijms-25-12175-f004:**
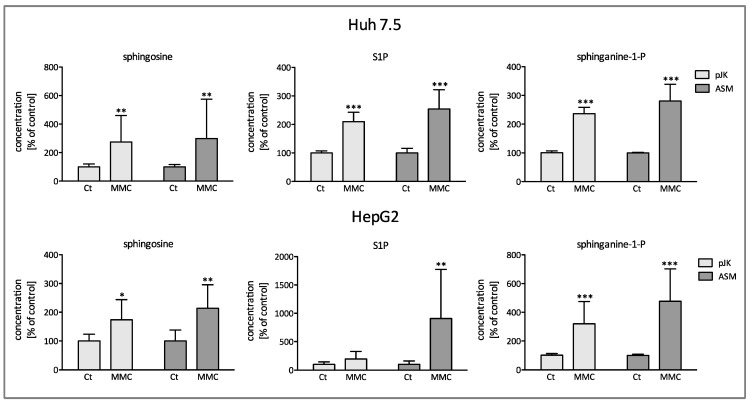
The effect of Mitomycin C on levels of sphingosine, S1P, and sphinganine-1-p. The ASM transfected (ASM) and control transfected (pJK) cells were treated for 48 h either with or without 3 µM mitomycin C (MMC). The results are expressed in % of the untreated control and are means ± SD (*n* = 3). *** *p* < 0.0001, ** *p* < 0.001, and * *p* < 0.05 indicate statistical significance when compared to the control group.

**Figure 5 ijms-25-12175-f005:**
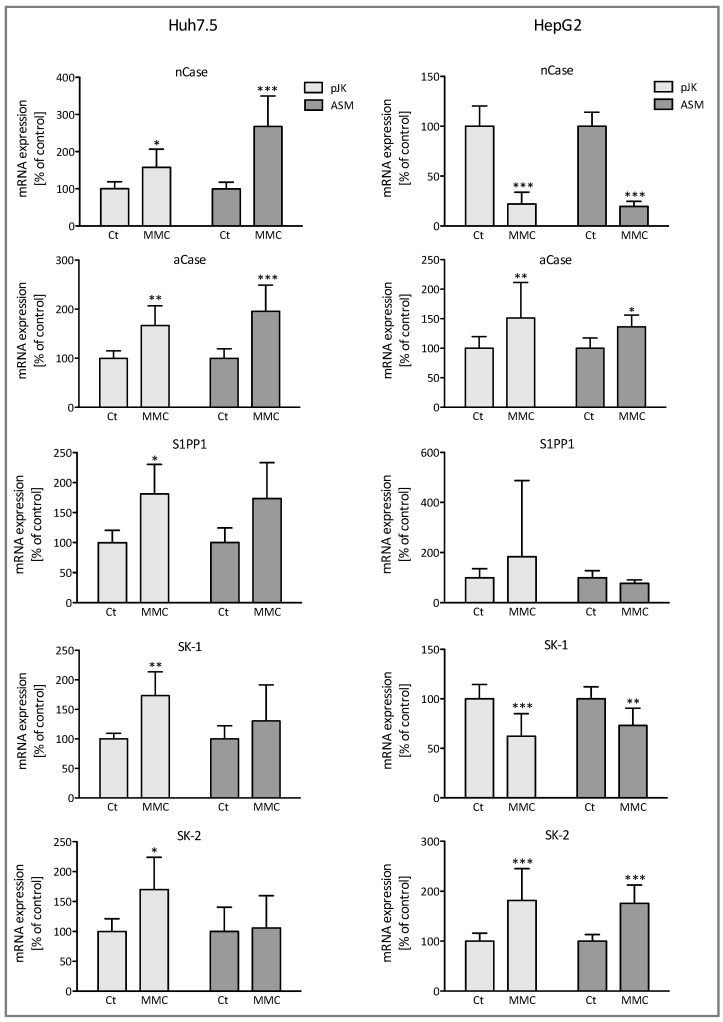
The mRNA expression of various enzymes of the sphingolipid metabolism upon Mitomycin C treatment. The ASM transfected (ASM) and control transfected (pJK) Huh7.5 und HepG2 cells were incubated for 48 h either with or without 3 µM mitomycin C (MMC). The data were obtained by the ΔΔCt method as described in the methods section and are expressed in % of untreated control, shown are means ± SD (*n* = 3). *** *p* < 0.0001, ** *p* < 0.001, and * *p* < 0.05 are considered significantly different when compared to the control group.

## Data Availability

The original contributions presented in the study are included in the article.

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
