# Peer review of "Acid Sphingomyelinase Activation and ROS Generation Potentiate Antiproliferative Effects of Mitomycin in HCC"

_ijms, 2024, doi:10.3390/ijms252212175_

Round 1
Reviewer 1 Report
Comments and Suggestions for Authors
please see my evaluation in the attached PDF file

Reviewer 2 Report
Comments and Suggestions for Authors
I found this article to be quite intriguing. While my expertise is primarily in lipid analysis via LC-MS, I believe the biological aspects are sound. To further enhance the manuscript, I suggest the following minor revisions:
The high standard deviation in MMC results warrants further discussion in the text. Why such a higher variability?
Include in the supplementary material a chromatogram of analysed samples and/or a table with monitored transitions and retention times.
Please adopt the current lipidomics nomenclature standards (see: 10.1194/jlr.S120001025)
Correct the labeling errors in Figure 3 and 4.
Consider using "Quantification via LC-MS" or simply "Quantification" instead of "mass spectrometric quantification".
Replace "Fig." with "Figure" in the discussion section.
